# Association of violence with urban points of interest

**Joseph Redfern**[1]*, **Kirill Sidorov**[1], **Paul L. Rosin**[1], **Padraig Corcoran**[1], **Simon C. Moore**[2], **David Marshall**[1]

**1** School of Computer Science and Informatics, Cardiff University, Cardiff, United Kingdom, **2** School of Dentistry, Cardiff University, Cardiff, United Kingdom

* RedfernJM@cardiff.ac.uk

**Data Availability Statement:** The primary data for this study comes from Police.uk (https://data. police.uk) and the Ordnance Survey via EDINA Digimap (https://digimap.edina.ac.uk/). Ordnance Survey data is available freely and easily for academic use via Digimap (https://digimap.edina.

## Abstract

The association between alcohol outlets and violence has long been recognised, and is commonly used to inform policing and licensing policies (such as staggered closing times and zoning). Less investigated, however, is the association between violent crime and other urban points of interest, which while associated with the city centre alcohol consumption economy, are not explicitly alcohol outlets. Here, machine learning (specifically, LASSO regression) is used to model the distribution of violent crime for the central 9 km$^2$ of ten large UK cities. Densities of 620 different Point of Interest types (sourced from Ordnance Survey) are used as predictors, with the 10 most explanatory variables being automatically selected for each city. Cross validation is used to test generalisability of each model. Results show that the inclusion of additional point of interest types produces a more accurate model, with significant increases in performance over a baseline univariate alcohol-outlet only model. Analysis of chosen variables for city-specific models shows potential candidates for new strategies on a per-city basis, with combined-model variables showing the general trend in POI/violence association across the UK. Although alcohol outlets remain the best individual predictor of violence, other points of interest should also be considered when modelling the distribution of violence in city centres. The presented method could be used to develop targeted, city-specific initiatives that go beyond alcohol outlets and also consider other locations.

## Introduction

Although there are many differing estimates as to the cost of alcohol-related crime to the police and health-care system, in the United Kingdom, it is widely accepted as being in the hundreds of millions of pounds per year [1]. In addition to the direct financial cost of alcohol related violence, there is a high societal cost—physical injuries sustained by victims can be life changing or fatal [2], the mental health of victims can be severely damaged [3], and reputations of cities can be permanently damaged [4]. A full understanding and quantification of how the distribution of violent crime varies is important when attempting to reduce the frequency of incidents, and could be useful in both preventative and reactive scenarios. Multiple studies examining

ac.uk/), and is available for evaluation under the Ordnance Survey data exploration license (https://www.ordnancesurvey.co.uk/business-government/licensing-agreements/data-exploration). However, commercial use of this third party data requires a paid license. Specifically, the study used the "Points of Interest" product from via Digimap, as described here: https://digimap.edina.ac.uk/webhelp/os/osdigimaphelp.htm#data_information/os_products/points_of_interest.htm. The authors had no special access to Police.uk or Ordnance Survey data.

**Funding:** Funding provided through Airbus Endeavr Wales/EADS Foundation Wales (https://airbusendeavr.wales/) grant named "Detecting deceit in humans through multimodal analysis of human behaviour". Funding awarded to DM/SCM/MI. SCM acknowledges the support of the Economic and Social Research Council, the Medical Research Council and Alcohol Research UK to the ELAStiC Project (ES/L015471/1). Funders did not play any part in the design, data collection/analysis, decision to publish, or preparation of the manuscript.

**Competing interests:** The authors have declared that no competing interests exist.

the relationship between the density of alcohol outlets and violent crime have been conducted, with clear relations being evident [5–7]. The results of these studies have important implications in city planning and policing. Typical "nights out" involve venues other than bars and clubs (such as fast-food outlets and public transport). The association between some of these other venues has been recognised [8, 9], but there have been limited attempts at determining exactly which venues contribute to violence, and to what degree. In this paper, we present an analysis of the relationship between various POI (point of interest) densities and levels of violent crime using machine-learning based approaches. We address some of the issues identified in existing literature by considering 10 different cities across the UK, building a regression model, and comparing our results to a univariate, alcohol-outlet only model. We show that considering multiple venue types results in a more accurate prediction of violent behaviour.

Violence is a broad term that includes reactive violence, typically in response to provocation, proactive violence that is premeditated and typically instrumental in achieving a predetermined goal, direct and indirect (where the intended victim is not directly harmed) violence [10]. The nature of violence referred to here is violence against the person, suggesting harm to the victim. Theoretical insights suggest several factors that interact to increase the likelihood of violence towards others in specific locations. Broadly, these factors are mostly associated with individual characteristics, such as personality, and environmental features, such as crowding and the action of others.

Across a population, the likelihood of violence will vary systematically. For example, an individual's experience of violence can, whether by direct experience or observational learning, increase the likelihood that they will become violent [10–12]. There are also strong socio-economic and demographic correlates with violence [13–15]. Male, socio-economically deprived individuals are more likely to be violent.

In addition, there are stable personality features, such as attention deficit disorder [16], that increase the likelihood of violence. However, stable personality features are mediated by the experience in a given time and place [17, 18]. Observing arousing events, including others' aggression, or experiencing stress can increase emotional arousal, which in turn increases the likelihood of violence [19, 20]. The implication being that features that are arousing or stressful can elicit emotional responses that increase the likelihood of violence. There are a number of features that might be expected to increase such a response and include undesired or uncontrollable events [20], competition when resources are scarce [21, 22], and overcrowding [23, 24], amongst others [25]. Furthermore, the initial response to these stressors is typically rapid, but baseline levels return over a longer period of time [26, 27], suggesting that individuals can remain aroused in a location and these levels of arousal will increase as the frequency of events increases.

One theory of violence, the I3 Model [25, 28, 29] provides further insights useful in understanding the relationship between environment and violence. According to this model, violence is determined by the interaction of three factors: (i) impulse (the situation or circumstances that trigger an aggressive response); (ii) impellence (the strength of the impulse for the individual); and (iii) inhibition (the strength of an inhibitory response overriding impulse to respond aggressively. Impulse recognises that people tend to become violent in response to certain types of events, such as provocation, goal obstruction and social rejection [25]. Inhibition is the ability to limit the effect of impulses on subsequent behaviour. Inhibition and impulse, consistent with Dual Systems Models [30], is regarded as being mediated by separate systems that develop at different rates. The rate of development for systems that mediate inhibition is slower than systems that process impulsive signals. This explains why impulsive behaviours, including violence, are most apparent in younger populations [31]. However, it also accounts for the relationship between alcohol consumption and violence. The

consumption of alcohol, and other psychoactive substances, is strongly implicated in violence [32]. It is argued that, pharmacologically, alcohol diminishes the capacity to inhibit impulses and therefore aggression [33]. Unsurprisingly, alcohol-related harms, including violence, have long been associated with premises licensed for the onsite sale and consumption of alcohol [34, 35]. However, alcohol is likely to have other effects beyond simply disinhibition [36]. Those who socialise in bars and other licensed premises will typically be younger and therefore more risk seeking [31] and the environment in and round premises will become crowded, noisy, leading to goal obstruction and other impellences such as unwanted social contact [37].

In summary, the proclivity for violence is systematically distributed across a population. Environments that attract demographics in which violence is more likely will exhibit an elevated level of violence. Moreover, locations that are stressful, whether that is due overcrowding or others' aggression, are likely to induce stress which in turn increases the likelihood of violence. Finally, locations in which alcohol can be consumed are more likely to see heightened levels of violence. However, many of these features are not unique to licensed premises suggesting that a narrow focus on such locations may fail to identify how the environment interacts with the individual to promote violence.

There are several works that model violence as a function of alcohol outlet density, with some examples including other variables such as deprivation levels and alcohol pricing.

Kinney et al. [38] studied the relationship between land use and urban crime rates in Burnaby, British Columbia, for both assaults and motor vehicle theft. Land use was split into categories (commercial, residential, and civic/institutional/recreational), and then further split into subcategories. It was found that the commercial land areas with the highest normalised assault rates were shopping centres, followed by pubs, hotels, and fast food restaurants. However, the work of Kinney et al. [38] leaves some potential areas of improvement. For example, of the commercial properties there were only 7 "regional shopping centers", despite a majority of assaults having been found to occur on this type of property. It is perhaps questionable to draw hard conclusions on the relationship between assault levels and shopping centres when there are so few instances—a single outlier could significantly impact these risk figures.

Nelson et al. [39] conducted a study of crime data in Cardiff and Worcester in which the locations of violent crimes were analysed and matched to types of venues. Their research showed that a majority of incidents occurred on roads, with some occurring in other areas (such as pubs, clubs, shops, car parks, and takeaways). Although insightful, the work did not involve correlation analysis or the formulation of a regression model, and the study was limited to two cities, making it difficult to determine whether their findings generalise to other cities or are limited to Cardiff and Worcester. Normalisation by total venue count was not applied, which somewhat limits the explanatory power of the analysis. For instance, if there are far more car parks in an area of analysis than there are cafes, but a marginally higher percentage of crimes were found to be committed in car parks compared to cafes, it cannot be concluded that there is a higher risk of violence associated with car parks.

Zhu et al. [7] analyse the relationship between alcohol outlet density and violence while controlling for neighbourhood socio-structural characteristics. The analysis was conducted over two Texan cities: Austin and San Antonio, using census data (188 and 263 tracts for each respective city). Ordinary Least Squares regression was applied to neighbourhood socio-structural data (primarily index of deprivation) and was found to explain 59% of violence for Austin and 39% of violence in San Antonio. Including alcohol outlet density further improved their model's performance. While the model of Zhu et al. [7] appears to perform well for the intended use case, it is less useful as a tool for city planning or policing on the scale that we consider in this paper. Use of census tracts limits the resolution of the model, particularly in city centre environments where residential density is low. In the UK, census areas are required

to contain at least 100 residents or 40 houses [40] and are built by merging postcode areas until this criterion is met, resulting in semi-arbitrarily shaped/sized areas. Even if census data was available on the address level, this may be of little use in the context of city centre drinking in the NTE (night-time economy), as only a minority of drinkers are likely to be city centre residents. Although many existing predictive crime models (such as Zhu et al. [7]) include census data, we chose to exclude this data from our research. Our model is designed to predict violence levels in city centres over a relatively small area, where in most cases a majority of revellers will have travelled in from residential areas—i.e, the census information is not relevant to non-residents.

## Materials and methods

### Materials

As input, our model uses the locations and types of POIs, as well as locations of reported incidents of crime. We perform our analysis using PointX POI data [41] and freely available police data [42]. These large datasets allow us to compare the effectiveness of our model across multiple cities while considering a number of different venue types.

**Open police data.**   Crime data sourced from *data.police.uk* [42] is provided after undergoing an anonymisation process, in which the location of the offence is quantised to either the centre of the road on which it took place, or to the nearest POI, as defined by Ordnance Survey and PointX respectively. The types of records available in the open police data [42] are summarised in Table 1. We restrict our analysis to incidents in the "Violence and sexual offences" crime type—sexual offences are inherently violent acts, broadly defined by rape and sexual assault. Such acts involve unwanted physical contact on the part of the victim, typically causing considerable physical and/or emotional harm, and as such, are included alongside other violent crimes.

Robberies (reports of which are ∼25 times less common than Violent and sexual offences in our dataset) were excluded from our analysis, as such offences are driven by different motivations than other violent crimes; they are acquisitive in nature, rather than being more spontaneous and reactionary (as discussed in Introduction).

In order to verify that the anonymisation process did not excessively skew the true distribution of crime and therefore impact our results, we conducted an analysis of the variation between open police data [42] locations and ground-truth crime locations. To this end, we obtained ground-truth non-anonymised crime data from South Wales Police (It was not practical to obtain such non-anonymised data from other police forces.) We computed the distance between true location and the closest anonymised location for each crime. The analysis of the resulting quantisation noise induced by anonymisation showed mean translation to be 31m,

**Table 1. Record fields, open police data [42].** Crime types are listed in S2 Appendix.

| NAME | DESCRIPTION |
|---|---|
| Crime ID | One-way hash of the record |
| Month | Month and year in which incident occurred |
| Reported by | Reporting police force |
| Falls within | Reporting police force |
| Longitude and latitude | Anonymised coordinates at which incident occurred |
| LSOA code and name | Census LSOA [42] name and code in which incident occured |
| Crime type | One of 14 crime types |
| Last outcome category | Outcome of incident, including status of investigation |

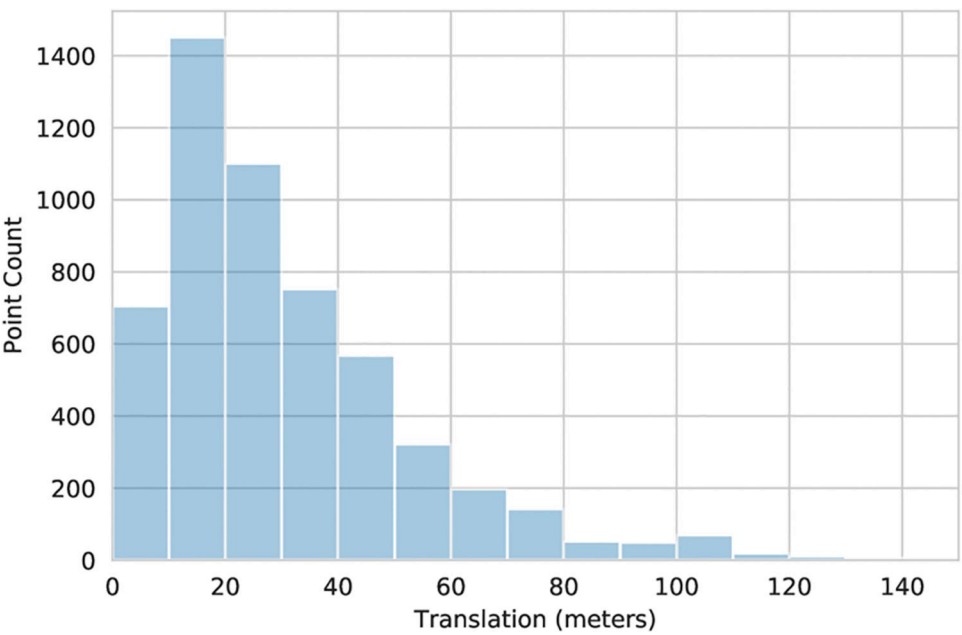

**Fig 1. Histogram of distances from anonymised to true crime locations (Cardiff).**

median translation 25m, with 90% of points having been translated by less than 60m. The histogram of quantisation noise is shown in Fig 1.

Furthermore, it should be noted that the temporal resolution of crime data is limited, due to aggregation on a monthly basis. This limits our ability to select crime occurrences that fall on a particular time of day. (For reference, we provide the plot of hourly violence reports by day of week for Cardiff in Fig 2.)

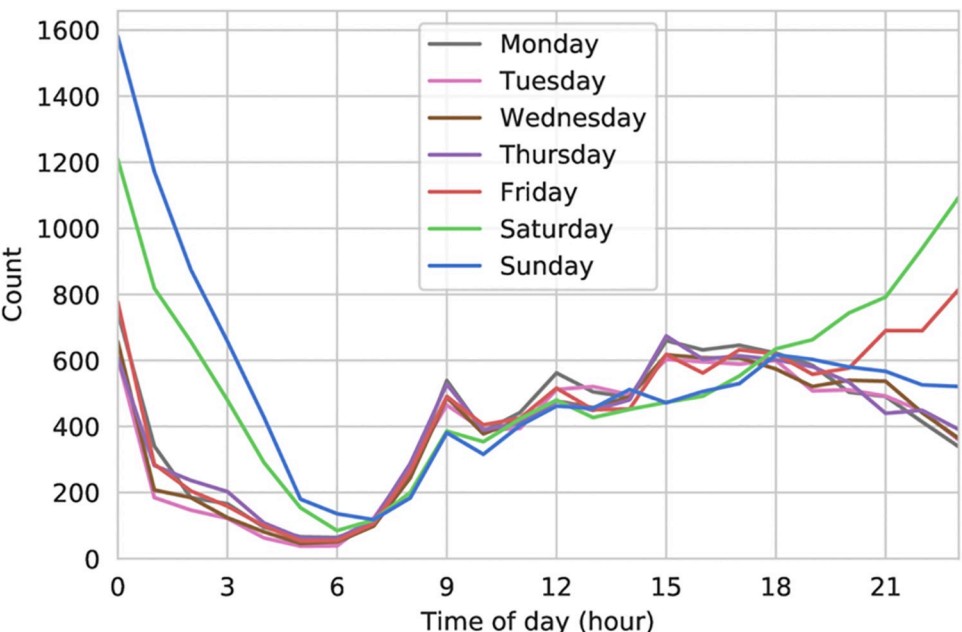

**Fig 2. Hourly reported violence count by day of week (Cardiff).**

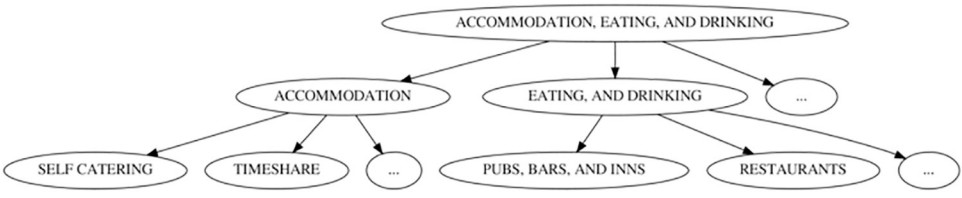

**Fig 3. Subset of point of interest hierarchy.**

**Point of interest data.** The point of interest data was created by PointX [41], licensed to Ordnance Survey, and obtained through EDINA Digimap [43]. The data includes the locations of every point of interest in the UK, grouped into 619 classes. The POI data fields include location (as an Ordnance Survey National Grid reference), name, and class.

The data is hierarchical in nature, with each class having a category, and each category in turn belonging to a group (as illustrated in Fig 3). Experimentally, we found that operating on the lowest level of the hierarchy yields the best results. The entire POI database contains over 4000000 points, and is described by PointX as "comprehensive" and "up-to-date", and is the basis of many mapping and satellite navigation systems.

This analysis considers the central 9 km$^2$ of the 10 most populated cities in England and Wales (listed in Table 2). Fig 4 thumbs illustrates these evaluation areas. We centred the 9 km$^2$ area on the centre of the city in question, the location of the city center being derived from Ordnance Survey map data. Crime data for Scotland is not available through Police.uk, so Edinburgh and Glasgow were excluded from the analysis. London was also excluded from the analysis, as we could not adequately define a single logical city centre.

## Methods

We assume that the effect of a point of interest on crime levels can be modelled as a function of Euclidean distance from it. (In reality, the situation may be more complicated, with anisotropic effects due to road network and other geospatial considerations.) We further assume that the total effect of multiple points of interest obeys the superposition principle: their effects are additive. In other words, the local crime density $C(\vec{x})$ at point $\vec{x}$ can be modeled as a linear combination of (possibly non-linear) effects due to points of interest $\vec{p}_i$:

$$C(\vec{x}) = \sum_i w_i K(D(\vec{p}_i, \vec{x})), \qquad (1)$$

**Table 2. Cities in the study and their populations, 2011 UK Census.** Note that population counts refer to entire city, $N_{\text{violence}}$ and $N_{\text{POI}}$ are the violence and POI counts in the central 9 km$^2$ of these cities.

| CITY | POPULATION | $N_{\text{violence}}$ | $N_{\text{POI}}$ |
|---|---|---|---|
| Birmingham | 1092330 | 2849 | 7680 |
| Leeds | 751485 | 3264 | 5963 |
| Sheffield | 552698 | 2122 | 4811 |
| Bradford | 522452 | 2752 | 3824 |
| Manchester | 503127 | 3960 | 7699 |
| Liverpool | 466415 | 2924 | 5178 |
| Bristol | 428234 | 3320 | 6031 |
| Cardiff | 346090 | 3178 | 4999 |
| Leicester | 329839 | 2804 | 5224 |
| Wakefield | 325837 | 1517 | 2657 |

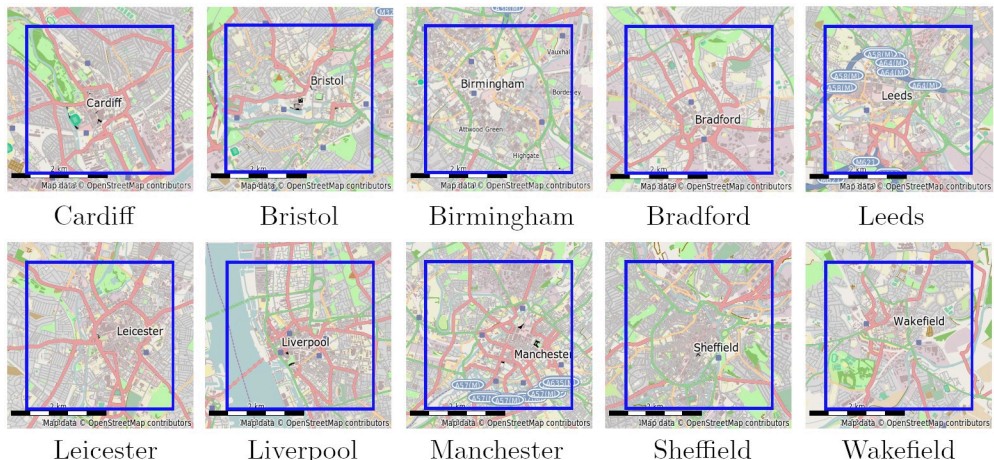

**Fig 4. City centres used for evaluation.** Blue squares indicate 3 km × 3 evaluation areas. © OpenStreetMap [44] contributors.

where $D$ is the distance function (in this paper, we use Euclidean distance, but other choices are possible such as network distance), $K$ is the kernel function (describing the decay of influence of the POI with distance), and $w_i$ is the weight for the $i$th POI. If we estimate the probability density function $C(\vec{x})$ from crime data, it is easy to estimate the model parameters $w_i$ by constructing and solving an over-determined linear system in the least-squared sense (for $M$ points of interest $\vec{p}_i$, and $N$ probe points $\vec{x}_j$ at which the crime density is known):

$$
\begin{cases}
w_1 K(D(\vec{p}_1, \vec{x}_1)) + \cdots + w_M K(D(\vec{p}_M, \vec{x}_1)) = C(\vec{x}_1) \\
\qquad\qquad\qquad\qquad \cdots \\
w_1 K(D(\vec{p}_1, \vec{x}_N)) + \cdots + w_M K(D(\vec{p}_M, \vec{x}_N)) = C(\vec{x}_N),
\end{cases}
\tag{2}
$$

or, in matrix form, $\vec{A}\vec{w} = \vec{c}$.

In this form, we assume that each individual POI (regardless of category) has its own weight. However, as a regularisation, we can assume the same weights $w_i$ apply to each POI category. By assuming POIs within each category have the same weight ($w_i = 1$), the only parameter of the model becomes POI locations—in this case, $w_i$ becomes a uniform scaling factor and can therefore be ignored. Under this assumption, the problem is then reduced to the problem of kernel density estimation. In the context of a regression problem, this results in a variable for each POI category, rather than one variable for each individual POI.

Our analysis operates on venue and crime densities, calculated from point data using kernel density estimation (KDE) [45]. Kernel density estimation allows us to estimate a continuous distribution from a set of discrete points, as a superposition of kernels centred at these points:

$$
F(\vec{x}) = \frac{1}{nh} \sum_{i=1}^{n} K\left( \frac{D(\vec{x}, \vec{p}_i)}{h} \right),
\tag{3}
$$

where as above, $D$ and $K$ are distance function and kernel respectively, and $h$ is the bandwidth (see discussion in *Kernel shape and bandwidth selection*). These density estimates can then be uniformly sampled, allowing us to compare densities at like-for-like positions in space. In order to prevent border effects, we apply padding (equal to the KDE bandwidth) when sampling the resulting kernel density estmation.

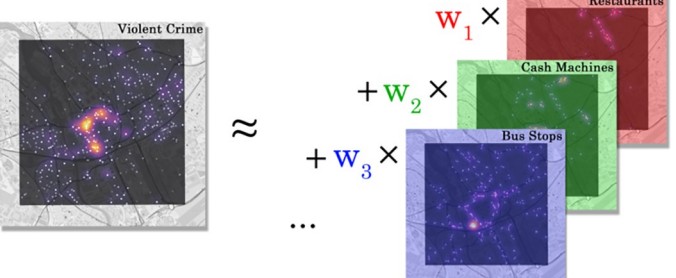

**Fig 5. Illustration of regression from POI densities to violent crime density.**

**Regression model.** We now attempt to predict crime level density by regression from densities of individual point of interest classes, with a variant of linear regression, as illustrated in Fig 5. More specifically, LASSO (least absolute shrinkage and selection operator) regression [46] was employed as our regression technique as it incorporates both regularisation and variable selection. Variable selection formed an important part of the model, as applying ordinary least squares (OLS) regression to all POI classes resulted in models which were overfit and difficult to interpret, and therefore did not significantly help in answering our research question. This was a result of collinearity between the distributions of many of the POI classes—for instance, clothing stores and footwear stores are often placed close to eachother. This collinearity would cause high model coefficients for one collinear variable, with low or even negative model coefficients for the remainder. This degree of collinearity is visualised in the correlation cluster map shown in Fig 6.

LASSO regression is similar to ridge regression [47] in that it includes variable regularisation (avoiding a bias towards variables with higher magnitudes), and adds an $L_1$ norm penalty term (weighted by a parameter $\alpha$) to variable coefficients. Unlike ridge regression, LASSO is able to *entirely* eliminate less significant variables from the model due to the coefficient penalty term being the $L_1$ norm rather than the $L_2$ norm of the variables' vector. More specifically, LASSO aims to minimise:

$$\frac{1}{2n}\|\vec{y} - \vec{B}\vec{w}\|_2^2 + \alpha\|\vec{w}\|_1 \tag{4}$$

In the above, $\vec{y}$ represents the anonymised vectorised crime density map, $\vec{w}$ represents the unknown mixture coefficients, $\vec{B}$ is a basis matrix whose columns are vectorised point of

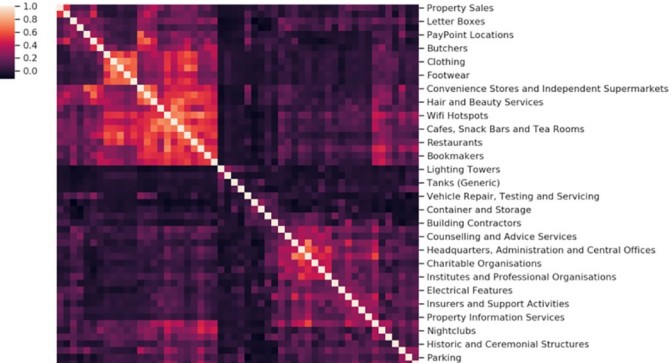

**Fig 6. Clustered correlation coefficients between subset of pairs of POIs (Cardiff).**

interest density maps, and $n$ is the number of samples (22500 in the case of a $150 \times 150$ density map). We derive the basis matrix $\vec{B}$ by first calculating kernel density estimates for each POI, using Eq (3). Each KDE is sampled over a regular 2D grid which is then flattened into a single vector, describing the spatial distribution of the POI. These vectors form the columns of the basis matrix.

We determine a value of $\alpha$ using binary search such that ten points of interest are selected (i.e. have non-zero coefficients). This is possible as LASSO is relatively computationally inexpensive and $\alpha$ is bound such that $0 \leq \alpha \leq 1$. In addition, we show the impact of varying $\alpha$ across a range of values, plotting its effect on both model performance, and non-zero variable count (see Fig 7).

Having determined ten most predictive points of interest with LASSO, we perform an ordinary least squares regressions using only these points of interest. Thus, we obtain a sparse linear regression model of crime density.

In addition to per-city models, we also developed a combined model, considering all ten cities simultaneously. This was achieved by merging all sampled city POIs and crime data (in consistent order) before performing LASSO. This combined model allows us to see the general trend across England and Wales, and can be used to help identify outlier cities with significantly different patterns of violence.

We establish a baseline score against which to compare our results by building linear regression models of alcohol outlet density alone to predict violence levels. Our baseline does not include census data, for reasons outlined in *Introduction*.

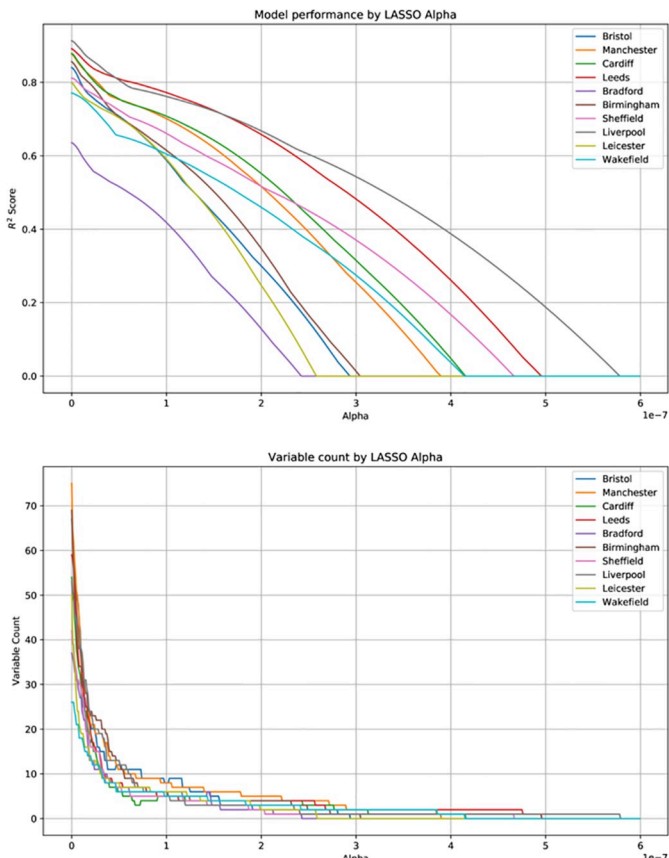

**Fig 7. Model performance and variable count by LASSO$\alpha$ value, all cities.**

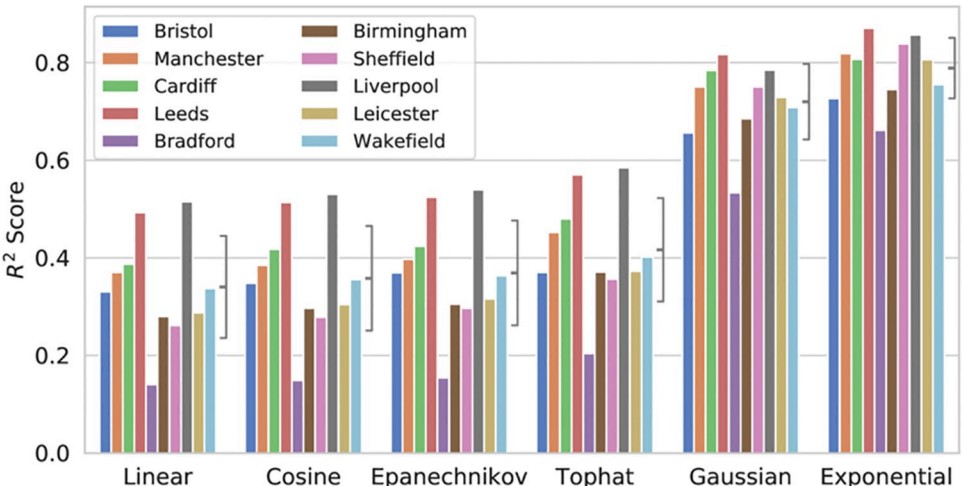

**Fig 8. Model performance by kernel type, brackets show mean and standard deviation in performance.**

**Kernel shape and bandwidth selection.** We now address the question of selecting the optimal kernel shape and bandwidth for kernel density estimation—these model the law according to which the influence of a POI decays with distance.

We evaluated the performance of multiple KDE kernels including Gaussian, top-hat, Epanechnikov, exponential, linear (triangular) and cosine. Contrary to some existing findings [48], for our data the best performing kernel was found to be the exponential kernel, beating all other kernels in overall model performance. The performance of these kernels is summarised in Fig 8.

Kernel bandwidth and cell-sizes for crime density estimates were selected in line with current advice—that is, using a cell size derived by dividing the shortest side of our study area by 150 [49], and a bandwidth derived by dividing the shortest side of our study area by between 30 and 50 [48] (we chose 40). As our study area size is fixed for all cities, this resulted in a consistent cell size and bandwidth of 20 m and 75 m respectively. Currently, the same bandwidth was chosen for our POI kernel density estimations. While the principle results in this paper are presented for this fixed crime KDE bandwidth, we conducted additional experiments to investigate the effect of variations in bandwidth choice on model performance. This is summarised in Fig 9. Note that while correlation scores seemingly improve with increasing bandwidth, the spatial resolution of the prediction is accordingly reduced. This bandwidth choice should thus be decided on a case-by-case basis, depending on the use case of the model.

## Results

We now present the results of applying the above methods to crime density predictions.

## City specific models

City-specific models were validated using K-fold cross validation, with $K = 2$. In each fold, 1/2 of the POI points in each class, and 1/2 of crime points were sampled uniformly randomly for the training set, with the remaining 1/2 set aside for testing. We report the results averaged over all folds, with this procedure further repeated 100 times with different random splits. (A value of $K = 2$ was chosen so as to preserve enough points in the test set for reliable density estimation.) The same process was used for our baseline model (alcohol outlets only). Example model output for Cardiff is shown in Fig 10.

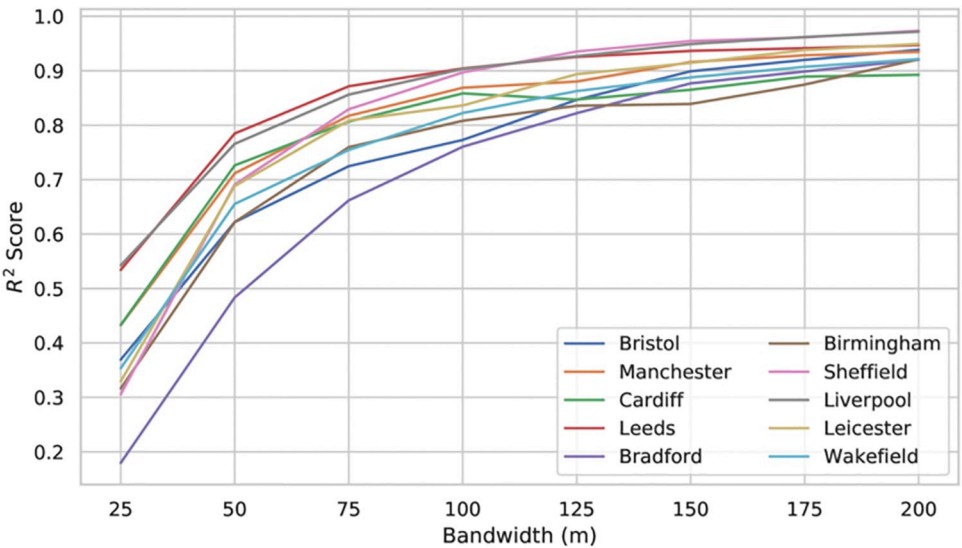

**Fig 9. The effect of varying the bandwidth in POI KDE, by city.**

We compare model performance using the coefficient of determination $R^2$, which measures how well our model is able to explain variance in our dependent variable (in this case, anonymised violent crime). $R^2$ scores are calculated between the ground-truth and predicted crime densities, uniformly sampled over the regular grid (as described in *Kernel shape and bandwidth selection*). City-specific model performance is presented in Table 3, with $R^2$ scores ranging from 0.43 to 0.71. The cities with best performing models were Leeds, Liverpool and Cardiff with scores of 0.71, 0.69, and 0.67 respectively. The worst performance was observed for Bradford, Wakefield, and Birmingham with scores of 0.43, 0.57, and 0.60 respectively.

It is clear that the relation between POI distribution and violence is variable across our evaluation areas. One possible explanation for this is due to environmental factors in the 3 km × 3 km regions. These regions were chosen based on the geographical center of the cities chosen for evaluation, as specified on Ordnance Survey maps. In some cases this will represent the entertainment/nightlife district of the city, however this is not guaranteed. It's likely that violence in residential areas has a stronger dependency on socioeconimic factors rather than POI distribution directly [50], so we can expect model performance to drop when our evaluation area overlaps with residential areas.

Across the 10 cities, our POI-based model offers *consistent improvements* compared to the baseline alcohol-only model, with increases in $R^2$ averaging at 43.5%, or 48.2% when excluding the potential outlier, Liverpool.

## Most significant points of interest

Counting the times each POI occurs in the top ten coefficients shows that alongside pubs, bars, and inns, there are several other POIs commonly associated with violence (listed in Table 4). Many of these POIs are logically associated with typical "nights out"—for instance, fast food and takeaway outlets may be frequented on the way home, bus stops may be used for transportation to and from drinkers' homes, and cash machines would commonly be visited over the course of an evening. We include coefficients for each city specific model in S1 Appendix.

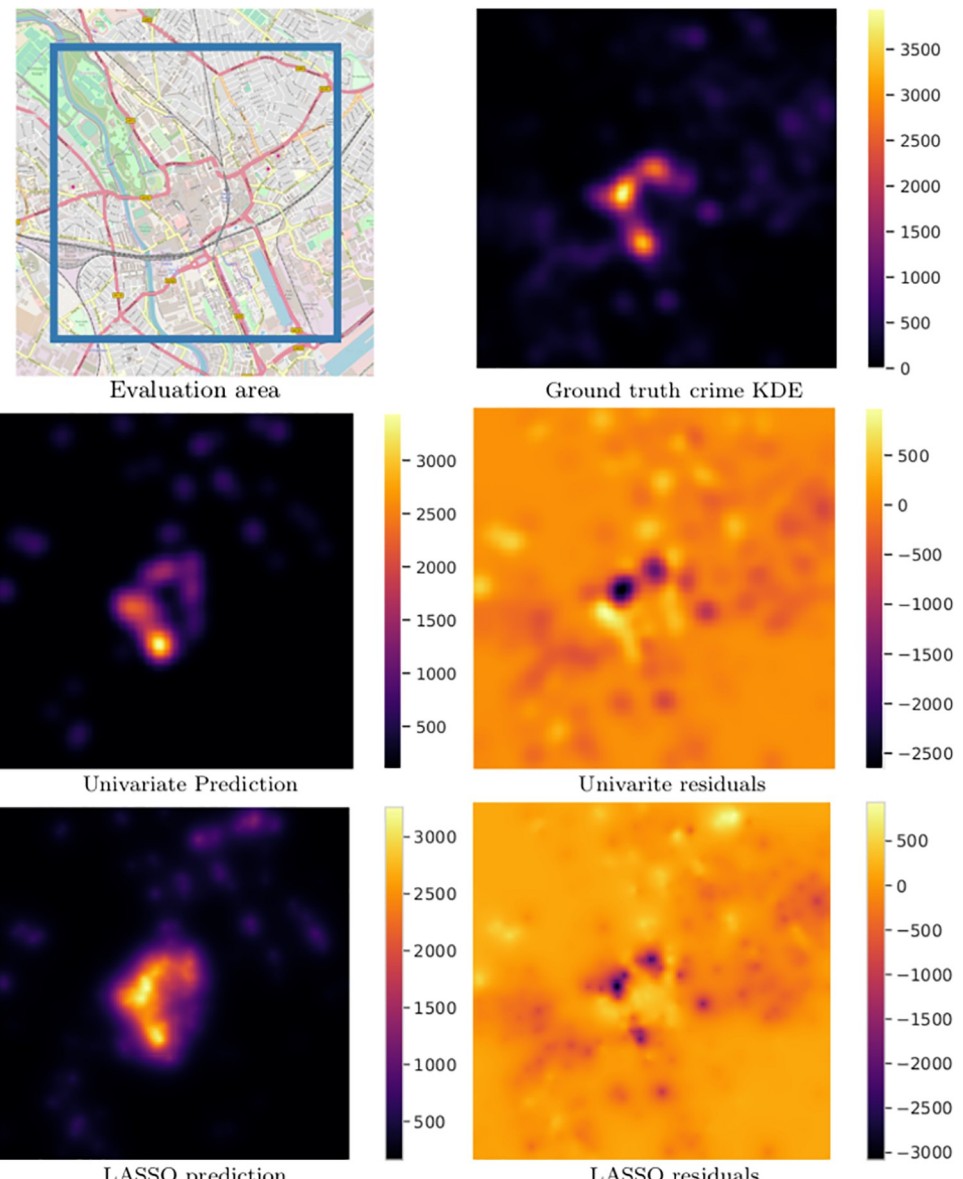

**Fig 10. Ground truth, prediction, and residuals for Cardiff.** Units are crime incidents per square kilometer.

## Combined model

We generated a combined model by merging the sampled points for all ten cities into a single vector, and merging the sampled crime KDEs for each city. We then performed LASSO regression on the merged data (see Table 5). In this experiment, we performed K-fold cross validation with $K = 10$, leaving out an entire city from each of the folds. This allows us to evaluate the model trained on nine cities against the tenth. While the K-fold validation in *City specific models* splits the POI and crime points within a city, thus the test and training set follow similar distributions, experiments in this section, leaving a city out entirely, are much more challenging as the distributions of POI and crime for the city under test are completely unknown during the training stage.

**Table 3. Mean performance of the baseline alcohol-outlet only linear regression vs the proposed LASSO-based POI model across 100 random test/train splits.**

| CITY | BASELINE | | PROPOSED MODEL | | IMPROVEMENT |
|---|---|---|---|---|---|
| | $R^2$ mean | st. dev. | $R^2$ mean | st. dev. | |
| Bristol | 0.2968 | 0.0476 | 0.6091 | 0.0439 | 105% |
| Manchester | 0.4405 | 0.0461 | 0.6443 | 0.0482 | 46% |
| Cardiff | 0.5632 | 0.0582 | 0.6729 | 0.0461 | 19% |
| Leeds | 0.6091 | 0.0552 | 0.7102 | 0.0500 | 16% |
| Bradford | 0.2681 | 0.0587 | 0.4309 | 0.0571 | 60% |
| Birmingham | 0.3754 | 0.0821 | 0.6033 | 0.0495 | 60% |
| Sheffield | 0.5220 | 0.0488 | 0.6295 | 0.0501 | 20% |
| Liverpool | 0.6825 | 0.0457 | 0.6926 | 0.0639 | 1% |
| Leicester | 0.3575 | 0.0354 | 0.6589 | 0.0321 | 84% |
| Wakefield | 0.4609 | 0.0787 | 0.5730 | 0.0763 | 24% |

**Table 4. Occurence count of POI classes occurring in top ten model coefficients.**

| POI CLASS | COUNT |
|---|---|
| Pubs, Bars, and Inns | 10 |
| Fast Food and Takeaway Outlets | 6 |
| Nightclubs | 6 |
| Bus Stops | 6 |
| Conv. Stores and Ind. Supermarkets | 5 |
| Charitable Organisations | 5 |
| Cash Machines | 4 |
| Counselling and Advice Services | 3 |
| Bookmakers | 3 |
| Hair and Beauty Services | 3 |

Perhaps unsurprisingly, the combined model has average performance, typically worse than the city-specific models (see comparison in Table 6). For certain cities, our combined model appears to out-perform the city-specific model. This is caused by the evaluation splits for some points of interests being too small in the city-specific model, and therefore not

**Table 5. Top 10 LASSO $R^2$ coefficients, combined model.**

| POI CLASS | $R^2$ COEFFICIENT |
|---|---|
| OVERALL $R^2$ | 0.5826 |
| Pubs, Bars and Inns | 0.1699 |
| Fast Food and Takeaway Outlets | 0.1031 |
| Bus Stops | 0.0980 |
| Nightclubs | 0.0555 |
| Cash Machines | 0.0521 |
| Conv. Stores and Ind. Supermarkets | 0.0143 |
| Counselling and Advice Services | 0.0083 |
| Restaurants | 0.0048 |
| Charitable Organisations | 0.0043 |
| Bookmakers | 0.0023 |

**Table 6. Combined model performance.**

| CITY | $R^2$ SCORE | |
|---|---|---|
| | City specific | Combined |
| Leeds | 0.7102 | 0.6831 |
| Leicester | 0.6589 | 0.5462 |
| Birmingham | 0.6033 | 0.6571 |
| Liverpool | 0.6926 | 0.6883 |
| Cardiff | 0.6729 | 0.6030 |
| Bristol | 0.6091 | 0.5386 |
| Manchester | 0.6443 | 0.4876 |
| Wakefield | 0.5730 | 0.0973 |
| Sheffield | 0.6295 | 0.4249 |
| Bradford | 0.4309 | 0.4184 |

capturing their true distributions. In the cases of Wakefield and Sheffield, the baseline model outperforms the combined model, suggesting that violence in the evaluation area for Liverpool is more strongly linked to alcohol outlets than other cities; or at least that it is less dependent on other points of interest.

Although city specific models that were trained on historical crime data for a single area offer the best performance, such models are not always practical. For example, while available in England and Wales, geographically detailed crime data is not necessarily made public in other countries, with only aggregate statistics being available for Northern Ireland and Scotland. In such cases, a model trained on other cities offers a good compromise between practicality and performance.

## Conclusion

Our work has identified several points of interest commonly associated with increases in violent crime, finding that many non-alcohol serving venues have a significant association with assault levels.

These findings are likely to have policy implications. Knowledge of which venues are likely to attract assaults is crucial when developing violence reductions strategies. Such strategies have been successfully introduced to alcohol outlets (for instance, staggered closing times and venue security requirements). Our work could help inform new, targeted initiatives which go beyond alcohol outlets, for instance around takeaways, public transport, and cash machines.

Although our combined model performs reasonably well, variations in the selected points of interest between different cities make it clear that an ideal violence reduction strategy would be developed on a city-by-city basis, and that without additional parameters, a generic "one-size fits all" approach is likely to be less effective.

Future studies may explore the possibility of custom kernels for different POI categories, in addition to the POI-specific kernel bandwidths already studied here. Use of non-Euclidean distance measures (such as network distance) in calculating densities may also be an interesting avenue of exploration, and could further improve results. Additionally, a more finely grained model may consider venue size, capacity, and interactions between venues, using this as the basis for non-uniform POI weights (see Methods). Finally, future work may further explore the possibility of sourcing data which includes time of incident, and only consider assaults that took place during the night.

## Supporting information

**S1 Appendix.**
(PDF)

**S2 Appendix.**
(PDF)

## Author Contributions

**Conceptualization:** Joseph Redfern, Kirill Sidorov, Paul L. Rosin, Padraig Corcoran, Simon C. Moore, David Marshall.

**Data curation:** Joseph Redfern.

**Funding acquisition:** Simon C. Moore, David Marshall.

**Investigation:** Joseph Redfern, Kirill Sidorov.

**Methodology:** Joseph Redfern, Kirill Sidorov, Paul L. Rosin, Padraig Corcoran.

**Project administration:** David Marshall.

**Software:** Joseph Redfern.

**Supervision:** Kirill Sidorov, Paul L. Rosin, Padraig Corcoran, Simon C. Moore, David Marshall.

**Visualization:** Joseph Redfern, Kirill Sidorov.

**Writing – original draft:** Joseph Redfern, Kirill Sidorov.

**Writing – review & editing:** Joseph Redfern, Kirill Sidorov, Paul L. Rosin, Padraig Corcoran, Simon C. Moore, David Marshall.

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
