## [Decision Letter · Decision Letter 0]

19 Mar 2020

PONE-D-20-03060

Association of violence with urban points of interest

PLOS ONE

Dear Mr Redfern,

Thank you for submitting your manuscript to PLOS ONE. After careful consideration, we feel that it has merit but does not fully meet PLOS ONE’s publication criteria as it currently stands. Therefore, we invite you to submit a revised version of the manuscript that addresses the points raised during the review process.

As your research is interdisciplinary, I have invited three qualified reviewers, an econometrician, a computational geographer, and an urban economist. Their recommendations are mixed. I myself do see some merits of your study by combining machine learning techniques to study the association between POI and crime. Please try to address the reviewers' concerns as much as you can.

We would appreciate receiving your revised manuscript by May 18. To enhance the reproducibility of your results, we recommend that if applicable you deposit your laboratory protocols in protocols.io, where a protocol can be assigned its own identifier (DOI) such that it can be cited independently in the future. For instructions see: http://journals.plos.org/plosone/s/submission-guidelines#loc-laboratory-protocols

We look forward to receiving your revised manuscript.

Kind regards,

Shihe Fu, Ph.D.

Academic Editor

PLOS ONE

Journal Requirements:

Reviewers' comments:

Reviewer's Responses to Questions

**Comments to the Author**

1. Is the manuscript technically sound, and do the data support the conclusions?

Reviewer #1: Partly

Reviewer #2: Yes

Reviewer #3: No

2. Has the statistical analysis been performed appropriately and rigorously? 

Reviewer #1: Yes

Reviewer #2: Yes

Reviewer #3: No

3. Have the authors made all data underlying the findings in their manuscript fully available?

Reviewer #1: Yes

Reviewer #2: No

Reviewer #3: Yes

4. Is the manuscript presented in an intelligible fashion and written in standard English?

Reviewer #1: Yes

Reviewer #2: Yes

Reviewer #3: No

5. Review Comments to the Author

Reviewer #1: 1. Equation 2 only describes one type of POI. There are N crime locations (\\vec{x_1} to \\vec{x_N}), M locations of point interests (\\vec{p_1} to \\vec{p_M}). However, the equation should also include different types of POIs, such as pubs, bus stops, supermarkets, etc., and the set of locations and weights are different for each type of POI.

2. The selection of the penalty term in Lasso regression should have theoretical justification. On page 6, it is stated that “We determine a value of \\alpha such that ten points of interest are selected”. Why only ten, given that there are 619 classes of POIs. Typically, for Lasso regression, one first determines a range for the penalty parameter, with the maximum being the smallest value such that no variable is selected. A grid of the penalty parameter is then constructed. The penalty term can be determined based on cross validation, looping over the possible grid points. One may also select the penalty term using plugin method based on theoretical justifications. The authors should provide some justification on the selection of the penalty term.

3. The paper considers the central 3kmx3km of cities, and density is estimated using univariate kernel density estimation. Note that for points on the boundary, only data points on one side are observed, which can lead to bias in density estimations. A solution may be to use a larger grid for density estimation and use the density estimates on a smaller grid to do the Lasso regressions.

4. The candidate variables in Lasso regression can include interactions between different classes of POIs. For example, the risk of crime may be high if alcohol outlets are mixed with cash machines in close distance.

5. On the first line on page 6, it is stated that “\\vec{y} represents the ground-truth vectorized crime density map”. I thought ground truth crime locations are only available for South Wales? (page 3, line 83)

6. How are the cities in the study selected? Some big cities are missing, such as London, Nottingham and Glasgow. The performance of the method in cities with more complicated structures is worth investigating.

7. Figure 10 (page 8) is missing from the manuscript.

8. The combined model performance (table 6) is worse than the baseline alcohol-outlet only model (table 3) for Liverpool. Some discussions can help clarify the issue.

Reviewer #2: Redfern et al investigated the association between violent crime and various urban points of interest. This is an interesting topic.

In the meantime, I have some comments and suggestions for the authors:

On Page 3 Table 1. “One of 15 crime types”

Crime Type: violent vs non-violent

It would be of interest to describe what are the 15 types of crimes the authors are referring to. Considering that the authors investigate the relationship between violent crime and points of interest, are all 15 types of crimes belong to the category of violent crime? Or some of them may belong to non-violent crimes (e.g. property crimes (such as theft), bribery, etc.)?

On Page 6 Line 157

Please give more details on how the authors get the ground-truth vectorized crime density map (y) and the basis matrix (B) since the definitions of them are missing. How the density of the violent crime was calculated at each point on the map from the locations of the offense in the open police data? I know that Eq (3) is based on the assumption that POIs within each category have the same weight. Eq(4) is the LASSO regression on the POI classes. Please give out the detailed derivation of the basis matrix (B) in Equation (4). It would be useful to give out the relationship of Eq(3) to the basis matrix (B) in the derivation.

On Page 7, Line 205, “Example model output for Cardiff is shown in Fig 9”

What are the values of the R-squared (in both Baseline model and Proposed model) for the example model output in Fig 9? Does higher R-squared always indicate a better prediction power in the Lasso based model (a higher level of similarity between Lasso predication and ground truth crime density) in authors' results?

On page 7, Line 207 and table 9

Definition is missing for R-squared score(s) authors put on line 207. How authors calculate the R-squared score? From Table 9, is R-squared score the same as the R-squared mean?

On page 7, Line 210

“with R2 scores ranging from 0.51 to 0.70”

If R-squared scores are defined as the same as R-squared mean in the authors’ manuscript, then the R-squared mean for the proposed model ranges from 0.52 to 0.65 (not from 0.51 to 0.70) based on the results from table 3. Where does this range '0.51 - 0.70' come from?

On page 8, Figure 10 missing

Figure 10 is missing. Please add figure 10.

Other Comments:

It is an interesting idea to utilize the density of points of interest in cities to study the violent crime density and to use the lasso regression to select the most important POI classes for each city. However, the violent crime density can be closely related to community factors of neighborhoods in a city (e.g. education level, median income level, etc). Various levels of community factors can play an important role on the quality of a POI. Due to this fact, the quality of each POI in the same POI class can still be very different from each other. This fact could be an important reason that the performance of the combined model (utilizing the other 9 cities' information and predicating the 10th city) displayed in table 6, is worse than the model for an individual city (except Birmingham).

Reviewer #3: This paper presents a study of the association between the density of different types of POIs and the density of violence crime counts. Though the topic itself would be interesting, I have several major concerns with the merit of this paper.

1. The contributions of this paper are not clear and convincing to me. First, the paper is not trying to contribute to the theories, as there is no discussion about the theories on what determine violence crime and the related literature. The interpretation of the results and the effects of different types of poi businesses is lacking and discussions in relation to relevant literature is missing. Second, the contribution to methodology is minor. Standard regression approach with variable selection is employed. The baseline model is too simple to make your main models good. The construction of poi related variables with kernel density was done a lot in the literature. Although the authors claim that one advantage of the work is to conduct the analysis at a fine spatial scale, the definition of unit of analysis is not clear to me. Third, regarding the empirical implication for policing and city-specific initiatives, why the found associations could be helpful and what reason or logical this conclusion was made are not explained and thus not convincing.

2. A comprehensive literature review regarding the theories on violence crime and related determining factors is needed. The current review is thin and mostly focuses on the method side.

3. The lack of details of models results and definition of variables. The dependent variable, density of crime counts, is not detailed. Coefficients of poi types for city-specific model are missing, which is critical for the explanation of why those factors vary across cities.

Given the above issues, I do not recommend the consideration of this paper for publication in this journal.

6. PLOS authors have the option to publish the peer review history of their article (what does this mean?). If published, this will include your full peer review and any attached files.

Reviewer #1: No

Reviewer #2: No

Reviewer #3: No

---

## [Author Response · Author response to Decision Letter 0]

11 Jun 2020

Thank you to both yourself and the reviewers for your feedback on our submission, entitled ``Association of violence with urban

points of interest''. We are glad the reviewers felt that this was an interesting topic. The comments were very useful and

constructive, and have resulted in what we believe is a significantly better piece of work.

I will now address the comments raised by the reviewers, and highlight changes made (where applicable). Line numbers refer to

those of the new, non-diffed paper. 

Firstly, \\textbf{Reviewer 1}:

\\begin{quotation}

 1. Equation 2 only describes one type of POI. There are N crime locations ($\\vec{x_1}$ to $\\vec{x_N}$), M locations of point interests ($\\vec{p_1}$ to $\\vec{p_M}$). However, the equation should also include different types of POIs, such as pubs, bus stops, supermarkets, etc., and the set of locations and weights are different for each type of POI.

\\end{quotation}

We agree that this may not have been clear enough, and have updated the text to clarify (L186). 

Under this equation, the point of interest type is not considered -- here we show a model that assumes each POI (regardless of

type) has its own weight. We then go on to state that we apply the same weight to each POI type (L188-189).

\\begin{quotation}

 2. The selection of the penalty term in Lasso regression should have theoretical justification. On page 6, it is stated that “We determine a value of $\\alpha$ such that ten points of interest are selected”. Why only ten, given that there are 619 classes of POIs. Typically, for Lasso regression, one first determines a range for the penalty parameter, with the maximum being the smallest value such that no variable is selected. A grid of the penalty parameter is then constructed. The penalty term can be determined based on cross validation, looping over the possible grid points. One may also select the penalty term using plugin method based on theoretical justifications. The authors should provide some justification on the selection of the penalty term.''

\\end{quotation}

%We agree that an analysis of the performance of the model over a range of regularsation values $\\alpha$ is warranted, and have added this to the paper (including plots showing the relation between $\\alpha$ and number of selected variables).

The number of predictors was limited to 10 arbitrarily, as a means of concisely summarising the N most important predictors across several cities.

For completeness, we have added an additional figure showing how $\\alpha$ effects both $R^2$ score and variable count (Figure 7).

\\begin{quotation}

 3. The paper considers the central 3kmx3km of cities, and density is estimated using univariate kernel density estimation. Note that for points on the boundary, only data points on one side are observed, which can lead to bias in density estimations. A solution may be to use a larger grid for density estimation and use the density estimates on a smaller grid to do the Lasso regressions.''

\\end{quotation}

We agree that the boundary condition should have been accounted for, and have now applied padding to the sample area.

We re-calculated our results and have updated the text (L199-200) to clarify this process. We also used this as an opportunity to adjust

number of folds ($K$) to 2, resulting in a more stable model (the addition of padding highlighted this issue).

\\begin{quotation}

 4. The candidate variables in Lasso regression can include interactions between different classes of POIs. For example, the risk of crime may be high if alcohol outlets are mixed with cash machines in close distance.''

\\end{quotation}

We agree that these interactions may result in a stronger predictive model, however believe that it may be outside the scope of

this particular study. We have updated the future work section to address this (L338), and plan to examine interactions as part

of our next work.

\\begin{quotation}

 5. On the first line on page 6, it is stated that ``vec{y} represents the ground-truth vectorized crime density map''. I

 thought ground truth crime locations are only available for South Wales? (page 3, line 83)''

\\end{quotation}

Reviewer 1 is correct, we have now clarified this (L220).

\\begin{quotation}

 6. How are the cities in the study selected? Some big cities are missing, such as London, Nottingham and Glasgow. The performance of the method in cities with more complicated structures is worth investigating.

\\end{quotation}

We agree that discussion around this was lacking.

Our study considers the 10 largest cities in England and Wales (excluding London, as we consider this an outlier). 

According to the statistics, Nottingham is not in the top 10 most populated cities in the UK (as it has a population of $305680$), so was excluded from the analysis.

Due to the population and scale of London, and the fact that we could not adequately define a ``center'' for the purposes of the study, we considered it an anomalous point so excluded it.

Scotland was excluded from the study as the same \\textit{Police.uk} crime data is not published by Police Scotland, making a like-for-like analysis impossible.

We have updated the text to clarify/state these points (L166-169)

\\begin{quotation}

 7. Figure 10 (page 8) is missing from the manuscript.''

\\end{quotation}

We have chosen to remove this figure; we believe a tabular representation is clearest (included in appendix)

\\begin{quotation}

 8. The combined model performance (table 6) is worse than the baseline alcohol-outlet only model (table 3) for Liverpool. Some discussions can help clarify the issue.''

\\end{quotation}

Having recalculated our results with padding and $K=2$, this is no longer the case for Liverpool -- however, it is for Sheffield

and Wakefield. We have added some discussion around this (L316-319).

Secondly, \\textbf{Reviewer 2}:

\\begin{quotation}

 1. It would be of interest to describe what are the 15 types of crimes the authors are referring to. Considering that the authors investigate the relationship between violent crime and points of interest, are all 15 types of crimes belong to the category of violent crime? Or some of them may belong to non-violent crimes (e.g. property crimes (such as theft), bribery, etc.)?

\\end{quotation}

This should have been made clearer. \\textit{Police.uk} crime data includes other crime types, such as burglary, shoplifting and

drugs-related incidents, which are not necessarily violent. We limit our analysis to one of these crime types, ``Violence and

Sexual offences''. We have updated the wording to make this clearer (L137-138), and have included a full list of crime types as

an appendix.

We have also rectified the crime category type count -- we had previously referenced 15 crime types, but this erroneously

included the pseudo-category "All crime". 

\\begin{quotation}

 2. Please give more details on how the authors get the ground-truth vectorized crime density map (y) and the basis matrix (B) since the definitions of them are missing. How the density of the violent crime was calculated at each point on the map from the locations of the offense in the open police data? I know that Eq (3) is based on the assumption that POIs within each category have the same weight. Eq(4) is the LASSO regression on the POI classes. Please give out the detailed derivation of the basis matrix (B) in Equation (4). It would be useful to give out the relationship of Eq(3) to the basis matrix (B) in the derivation.

\\end{quotation}

We have now attempted to clarify that the basis matrix B is composed of sampled POI kernel density estimates, and hope that this

clarifies the process (L223-226).

\\begin{quotation}

 3. What are the values of the R-squared (in both Baseline model and Proposed model) for the example model output in Fig 9? Does higher R-squared always indicate a better prediction power in the Lasso based model (a higher level of similarity between Lasso predication and ground truth crime density) in authors' results?

\\end{quotation}

$R^2$ values for the Figure 9 are listed in Table 3.

We have attempted to clarify that $R^2$ is indicative of better model performance (L275-277)

\\begin{quotation}

 4. Definition is missing for R-squared score(s) authors put on line 207. How authors calculate the R-squared score? From Table 9, is R-squared score the same as the R-squared mean?

\\end{quotation}

As above, we have now better defined $R^2$ (L275-277).

As for $R^2$ mean and standard deviation, we agree that this should be more clearly defined, and have done so in the revised

paper (Table 3 caption)

\\begin{quotation}

 5. ``with R2 scores ranging from 0.51 to 0.70''

If R-squared scores are defined as the same as R-squared mean in the authors’ manuscript, then the R-squared mean for the proposed model ranges from 0.52 to 0.65 (not from 0.51 to 0.70) based on the results from table 3. Where does this range '0.51 - 0.70' come from?

\\end{quotation}

Reviewer 2 is correct, we had inadvertantly reported the range over the folds, rather than range of the mean of the folds.

This has now been corrected (L280).

\\begin{quotation}

 6. On page 8, Figure 10 missing

 Figure 10 is missing. Please add figure 10.

\\end{quotation}

As above -- we have now chosen to remove what was figure 10, as we believe tabular formatting is the clearest representation.

\\begin{quotation}

 7. It is an interesting idea to utilize the density of points of interest in cities to study the violent crime density and to

 use the lasso regression to select the most important POI classes for each city. However, the violent crime density can be

 closely related to community factors of neighborhoods in a city (e.g. education level, median income level, etc). Various

 levels of community factors can play an important role on the quality of a POI. Due to this fact, the quality of each POI in

 the same POI class can still be very different from each other. This fact could be an important reason that the performance

 of the combined model (utilizing the other 9 cities' information and predicating the 10th city) displayed in table 6, is

 worse than the model for an individual city (except Birmingham).

\\end{quotation}

We fully agree that in general, models of violent crime benefit from the inclusion of census data, for the reasons you outline.

However, we are not convinced of the value gained by including this data in this specific analysis.

Firstly, our model focuses on city centres, where permenant residents typically make up a minority of occupants. Census data in

our evaluation area unlikely to be reflective of a majority of visitors to city centres, and so without sigificant changes

(for instance considering residential areas neighbouring the study area) risks adding confusion to the model.

Secondly, it is difficult to act upon knowledge that education level or income levels are associated with increases in violence.

It is much easier to act on knowledge that, for instance, your city tends to see an increase in assaults around a particular

venue type -- this can be acted upon immediately, rather than requiring years (or decades) of fundemental changes to schooling

and socioeconomic policy.

This is not to say that we disagree that individual points of interest within the same category may have stronger or weaker

contributions to violence levels, this is definitely the case. In future work will look to integrate additional variables,

such as venue opening hours, venue size and average unit pricing -- but we believe that the analysis as it stands is still very

useful.

Finally, \\textbf{Reviewer 3}:

\\begin{quotation}

 1. First, the paper is not trying to contribute to the theories, as there is no discussion about the theories on what determine violence crime and the related literature. The interpretation of the results and the effects of different types of poi businesses is lacking and discussions in relation to relevant literature is missing.

\\end{quotation}

We agree that the discussion around theories of violent crime was lacking, and have now included a more extensive discussion and literature review (L23-79).

\\begin{quotation}

 2. Second, the contribution to methodology is minor. Standard regression approach with variable selection is employed.

 The baseline model is too simple to make your main models good. The construction of poi related variables with kernel density was done a lot in the literature.

 Although the authors claim that one advantage of the work is to conduct the analysis at a fine spatial scale, the definition of unit of analysis is not clear to me. 

\\end{quotation}

This is an interdisciplinary study which discusses a new approach to relating city-centre features to crime levels.

In the context of this work, we believe that the use of standard techniques is a positive rather than a negative. We are using proven techniques

to conduct a novel analysis. We found no other literature that studied such a broad range of points of interest over multiple cities. If this is

not the case, and this analysis has been conducted before, then it would be useful to be provided with a reference to the work.

The use of Kernel Density Estimates for crime prediction is common in the literature, which we have cited. As for our model baseline, we believe

this to be justified. For policy making, the locations of POIs such as pubs, takeaways and cash machines are something that is directly controllable

through i.e. planning permission decisions. Other models typically consider census-derived variables, such as income or age, which are far harder to

influence through policy decisions.

\\begin{quotation}

 3. A comprehensive literature review regarding the theories on violence crime and related determining factors is needed. The current review is thin and mostly focuses on the method side.

\\end{quotation}

As above, we agree that the initial literature review was too thin, and have now significantly expanded upon it, with a stronger focus on theories behind violent crime (L23-79)

\\begin{quotation}

 4. The lack of details of models results and definition of variables. The dependent variable, density of crime counts, is not detailed. Coefficients of poi types for city-specific model are missing, which is critical for the explanation of why those factors vary across cities.

\\end{quotation}

In hindsight, this was a definite mistake. We have now included model coefficients for each city as an appendix (appendix 1).

Yours Sincerely,

Joseph Redfern

---

## [Decision Letter · Decision Letter 1]

6 Aug 2020

PONE-D-20-03060R1

Association of violence with urban points of interest

PLOS ONE

Dear Dr. Redfern,

Thank you for submitting your manuscript to PLOS ONE. After careful consideration, we feel that it has merit but does not fully meet PLOS ONE’s publication criteria as it currently stands. Therefore, we invite you to submit a revised version of the manuscript that addresses the points raised during the review process.

Both reviewers still have concerns. Reviewer 2 raised the question whether sexual offense is violent or not. You may consider replacing "violence" with "crime" in the title if you cannot clarify this. Please try to address their other conerns as much as you can.

We look forward to receiving your revised manuscript.

Kind regards,

Shihe Fu, Ph.D.

Academic Editor

PLOS ONE

Reviewers' comments:

Reviewer's Responses to Questions

**Comments to the Author**

1. If the authors have adequately addressed your comments raised in a previous round of review and you feel that this manuscript is now acceptable for publication, you may indicate that here to bypass the “Comments to the Author” section, enter your conflict of interest statement in the “Confidential to Editor” section, and submit your "Accept" recommendation.

Reviewer #1: (No Response)

Reviewer #2: (No Response)

2. Is the manuscript technically sound, and do the data support the conclusions?

Reviewer #1: Yes

Reviewer #2: Partly

3. Has the statistical analysis been performed appropriately and rigorously? 

Reviewer #1: Yes

Reviewer #2: N/A

4. Have the authors made all data underlying the findings in their manuscript fully available?

Reviewer #1: Yes

Reviewer #2: Yes

5. Is the manuscript presented in an intelligible fashion and written in standard English?

Reviewer #1: Yes

Reviewer #2: Yes

6. Review Comments to the Author

Reviewer #1: Thanks for revising the paper. I have only a minor concern regarding the response to my previous comment 1.

In the Methods section, I think equations 1 and 2 should be dropped.

The analysis begins with estimating kernel densities of crimes and each POI separately using equation 3, which has nothing to do with explaining the crime density by POIs. Then with the estimated densities, Lasso regression is performed to predict the crime density using the densities of POIs (equation 4). In this line of reasoning, there is no need to assume “POIs within each category have the same weight” (line 188, page 6). Please clarify if my interpretation is not right.

Reviewer #2: I appreciate for your revision. Thanks for explaining your data categories to me, which addresses some of my concerns – the crime type you use, and which let me know the details of the crime categories through the added appendix 2. I know you are investigating the relationship of violence with POI. However, after look into the details of the crime categories, you provided in the revision, I do not think (at least at this point) the database you use is persuasive to me, and therefore, I am recommending a further revision. First of all, you choose to use violence and sex offense. But is this category of a crime necessary to belong to violent crime? Sex offense may or may not involve violence. Secondly, from the other crime types you provide in the revision (Table 1 in appendix 2), how can you tell the other crime category is not belong to violent crime? I think it is worth investigating which crime type is exactly violent which is not? At least, I can tell that Robbery belongs to violent crime and it is the type of crime that is available in the dataset, but you did not include it in your study. Are you certain that sex offense in the database is all belong to violent sex offense? I also suggest you add other violent crime categories (e.g. Robbery and other violent categories) into your data and to perform the analysis.

In respect to the combined model, I think it is meaningless because every city is different, (such as income level, educational level). I do not find that it would be useful to use a model trained from other cities to perform predication in another city. The database you have is only about the POI in a very general way, but we know that even a restaurant, there is a high ending one, and there is so-so one, which depends on the income level of a city. I think those differences have an impact on violence.

7. PLOS authors have the option to publish the peer review history of their article (what does this mean?). If published, this will include your full peer review and any attached files.

Reviewer #1: No

Reviewer #2: No

---

## [Author Response · Author response to Decision Letter 1]

11 Sep 2020

Dear Editor,

Thank you for arranging the second review of our submission, entitled “Association of violence with urban points of interest”. We appreciate the additional constructive comments from both reviewers, which we believe have once again resulted in an improved manuscript.

As before, I will now address the comments raised by the reviewers, and highlight changes made (where applicable). Line numbers refer to those of the new, non-diffed paper.

Firstly, Reviewer 1:

> In the Methods section, I think equations 1 and 2 should be dropped. The analysis begins with estimating kernel densities of crimes and each POI separately using equation 3, which has nothing to do with explaining the crime density by POIs. Then with the estimated densities, Lasso regression is performed to predict the crime density using the densities of POIs (equation 4). In this line of reasoning, there is no need to assume “POIs within each category have the same weight” (line 188, page 6). Please clarify if my interpretation is not right.

Equations 1 and 2 show a model that assumes that each point of interest is independently associated with crime levels, with each individual POI having its own weighting. In the context of a regression model, this is equivalent to having one independent variable for each point of interest. We attempted to explain (on line 187) that as a regularisation, we use the same weight for each POI within the same category – i.e. that all POIs of type “Cafe” would have weight x cafe , that all POIs of type “Pub” would have weight x pub . In this case, in the context of a regression model, this is equivalent to having one independent variable for each point of interest category.

We include equations 1 and 2 to try and justify and explain how we arrived at equation 3. Unlike the non-regularised model, Equation 3 does assume that POIs within the same category have the same weight in our model. 

With this in mind, we believe that the inclusion of Equations 1 and 2 are justified; however we have attempted to clarify this point by updating the text (L199–201).

Secondly, Reviewer 2:

> However, after look into the details of the crime categories, you provided in the revision, I do not think (at least at this point) the database you use is persuasive to me, and therefore, I am recommending a further revision. First of all, you choose to use violence and sex offense. But is this category of a crime necessary to belong to violent crime? Sex offense may or may not involve violence.

We have no control over how categories are allocated to crime reports. “Violent and Sexual Offences” exists as a single stand alone category, rather than two categories “Violent” and “Sexual”.

However, even if such a distinction were possible, we would be inclined to include Sexual offences in our analysis anyway. A sexual offence is a form of violence of a sexual nature, broadly defined by rape and sexual

assault. It involves unwanted physical contact, on the part of the victim, and typically causes considerable physical and/or emotional harm.

We have updated the text to clarify this point (L139–142).

> Secondly, from the other crime types you provide in the revision (Table 1 in appendix 2), how can you tell the other crime category is not belong to violent crime? I think it is worth investigating which crime type is exactly violent which is not? At least, I can tell that Robbery belongs to violent crime and it is the type of crime that is available in the dataset, but you did not include it in your study. Are you certain that sex offense in the database is all belong to violent sex offense? I also suggest you add other violent crime categories (e.g. Robbery and other violent categories) into your data and to perform the analysis.

It is true that *certain* robberies could be considered violent acts, however, they are not necessarily violent. Our concern is that incorporating this crime type into our model may introduce as many false negatives as

it would additional true positives.

The data needed to ascertain which of those robberies were violent and which were not violent is not in the public domain, so such an investigation, while interesting, is unfortunately not possible.

Following your suggestions, we investigated how many crime reports were allocated the crime type “Violence and Sexual Offences” and how many were allocated the crime type ”Robbery”. There were 423, 624 instances of “Violence and Sexual Offences”, and 18, 119 instances of “Robbery”, i.e. “Violence and Sexual offences” were 25 times more common. If 50% of Robbery involved violence, exclusion of the category would represent a loss of ≈2% under-reporting of violent incidents.

Furthermore, the motivation behind a robbery is different to that of violence. Robberies are typically planned and are acquisitive — i.e. the intention is to acquire goods (albeit under the threat of physical harm). However, other violence is typically more spontaneous and reactionary.

It is worth noting that these crime types are determined by how the crime was reported to the police. If the main concern was due to violence, the report may have been recorded as a violent crime. If the main concern was the (attempted) theft of property, the crime may have been recorded as robbery. With this in mind, we are confident that considering “Violent and Sexual incidents” alone is the right decision.

We have updated the text to justify the exclusion of robberies from our analysis (L143–146).

> In respect to the combined model, I think it is meaningless because every city is different, (such as income level, educational level). I do not find that it would be useful to use a model trained from other cities to perform predication in another city. The database you have is only about the POI in a very general way, but we know that even a restaurant, there is a high ending one, and there is so-so one, which depends on the income level of a city. I think those differences have an impact on violence.

Although we appreciate that a dedicated, city-specific model is optimal in most cases, we disagree that a combined model is meaningless.

As described in the manuscript (L307), we train the combined model on 9 cities, and evaluate against the remaining one, repeating 10 times, reporting our results. If this process gave 0 or negative R 2 values, then the model would be useless. However, we achieve R 2 > 0.4 in almost every case. This makes the combined model useful for areas without public crime data (for instance, in Scotland or Northern Ireland, which are not part of the Police.uk dataset). Every city is different, but not different enough that a combined model is useless. Taking into account the quality or clientele of individual POIs within the same category, while an interesting idea, is beyond the scope of this work.

We have updated the text to clarify the use case of the combined model (L331–336).

Thank you for considering our revised manuscript, we hope we have fully addressed and clarified the remaining concerns of the reviewers.

Yours sincerely,

Joseph Redfern (corresponding author)

Email: RedfernJM@cardiff.ac.uk

---

## [Editor Report · Decision Letter 2]

15 Sep 2020

Association of violence with urban points of interest

PONE-D-20-03060R2

Dear Dr. Redfern,

We’re pleased to inform you that your manuscript has been judged scientifically suitable for publication and will be formally accepted for publication once it meets all outstanding technical requirements.

Kind regards,

Shihe Fu, Ph.D.

Academic Editor

PLOS ONE
---

## [Editor Report · Acceptance letter]

16 Sep 2020

PONE-D-20-03060R2 

Association of violence with urban points of interest 

Dear Dr. Redfern:

I'm pleased to inform you that your manuscript has been deemed suitable for publication in PLOS ONE. Congratulations! Your manuscript is now with our production department. 

Kind regards, 

on behalf of

Dr. Shihe Fu 

Academic Editor

PLOS ONE